# Energy Efficient Routing Algorithm with Mobile Sink Support for Wireless Sensor Networks

**DOI:** 10.3390/s19071494

**Published:** 2019-03-27

**Authors:** Jin Wang, Yu Gao, Wei Liu, Arun Kumar Sangaiah, Hye-Jin Kim

**Affiliations:** 1Hunan Provincial Key Laboratory of Intelligent Processing of Big Data on Transportation, School of Computer & Communication Engineering, Changsha University of Science & Technology, Changsha 410000, China; jinwang@csust.edu.cn; 2College of Information Engineering, Yangzhou University, Yangzhou 225000, China; gaoyuyz@163.com (Y.G.); yzliuwei@126.com (W.L.); 3School of Information Science and Engineering, Fujian University of Technology, Fuzhou 350000, China; 4School of Computing Science and Engineering, Vellore Institute of Technology (VIT), Tamil Nadu 632014, India; sarunkumar@vit.ac.in; 5Business Administration Research Institute, Sungshin W. University, Seoul 100744, Korea

**Keywords:** Wireless sensor network (WSN), clustering, mobile sink, moving trajectory scheduling, energy efficiency

## Abstract

Recently, wireless sensor network (WSN) has drawn wide attention. It can be viewed as a network with lots of sensors that are autonomously organized and cooperate with each other to collect, process, and transmit data around targets to some remote administrative center. As such, sensors may be deployed in harsh environments where it is impossible for battery replacement. Therefore, energy efficient routing is crucial for applications that introduce WSNs. In this paper, we present an energy efficient routing schema combined with clustering and sink mobility technology. We first divide the whole sensor field into sectors and each sector elects a Cluster Head (CH) by calculating its members’ weight. Member nodes calculate energy consumption of different routing paths to choose the optimal scenario. Then CHs are connected into a chain using the greedy algorithm for intercluster communication. Simulation results prove the presented schema outperforms some similar work such as Cluster-Chain Mobile Agent Routing (CCMAR) and Energy-efficient Cluster-based Dynamic Routing Algorithm (ECDRA). Additionally, we explore the influence of different network parameters on the performance of the network and further enhance its performance.

## 1. Introduction

Recent advances in wireless communication, and computer and micro-electronic technology have enabled the rapid development of tiny, low-cost, and multi-functional sensors. These sensors are commonly deployed in the target areas via a random way to monitor the physical features of the surroundings such as temperature, humidity, and pressure. The monitored data is usually forwarded to the data collector (sink) using a cooperative method (generally multi-hop) and the collector may relay the data to a remote server for further data analysis. Besides, sensors can self-organize themselves based on their local collaboration to form wireless sensor networks (WSNs) [1,2,3]. The favorable characteristics of WSNs, such as rapid deployment, high fault tolerance, self-organization, real-time data transport, etc., make them suitable to be deployed in unfriendly, or even harsh environments, especially in military or disaster surveillance. Additionally, WSNs are also extensively applied in industrial product line monitoring, agricultural and wildlife observation, healthcare, smart homes, etc. [4,5].

Sensors are generally powered by energy-constrained batteries and it is unrealistic for battery replacement because of the enormous quantity of sensors and expensive cost. The best solution for the insufficient energy in WSNs is to equip the sensors with higher-capacity batteries. However, the performance of the battery has reached its choke point and is hard to be improved [6,7]. Therefore, the energy shortage problem in WSNs could only be addressed by adopting energy efficient routing protocols or algorithms [8,9]. Additionally, WSNs are also troubled with the unbalanced energy of sensors. Each sensor owns its monitoring range and once a node dies, fade zones will appear and the performance of the network will also rapidly decrease. The phenomenon of unbalanced energy in WSNs is similar to the energy holes, and they are both caused by the “hot spots” problem. The “hot spots” problem usually happens in WSNs with a static sink and fixed network topology. Nodes around the sink are usually burdened with busier traffic because they need to transfer the data packages from the outer layer to the sink and exhaust their energy faster. Since the sink is static and the network topology remains unchanged, the energy becomes more and more heterogeneous as time goes on. Network lifetime is an important evaluation criterion to estimate the performance of the network and it is commonly defined as the time when the first node dies. 

A great deal of work has focused on the problem of energy efficiency and energy balance, and great results have been achieved [10,11,12,13]. Clustering technology greatly decreases the energy consumption of WSNs by dividing the sensors into clusters according to some special rules. In each cluster, one or more cluster heads (CHs) are elected to be the relay nodes for its members. Clustering helps the network simplify the topology structure and avoids the direct communication between sensors and the sink. Additionally, data fusion can be adopted in CHs to filter the redundant data to decrease the burden of CHs. One of the classic routing protocols that adopts clustering is the low-energy adaptive clustering algorithm (LEACH); however, the approach for CHs selection is unreasonable and much further work is being conducted based on LEACH. 

Sink mobility technology is an efficient method to deal with the unbalanced energy in WSNs. In mobile sink-supported WSNs, the sink is usually carried by the intelligent vehicles or robots and it can move freely around the sensing field. The following advantages can be achieved by introducing sink mobility technology. First, the “hot spots” problem can be greatly alleviated by the movement of the sink. Areas around the sink are usually traffic hubs and when the sink moves, the traffic hub also transfers. Sensors take turns to be the “hot spots” to evenly distribute the energy consumption. Second, the total energy consumption can be greatly reduced by abridging the transmission distance between the communication pairs, assuming that the sink mobility pattern is well designed. Third, the latency of the network can be reduced and the throughput of the network can be increased by using a mobile sink. Finally, network connectivity can be ensured, even under the sparse or disconnected sensor networks. Although the introduction of sink mobility involves many merits, it also faces some challenges [14]. The location of the mobile sink should be broadcast frequently or predicted by sensors, which may increase the burden of the network. Additionally, the moving pattern of the sink should be well designed to cooperate with the local nodes for data transmission.

For addressing the above problems, in this paper, we present an energy-efficient routing schema combined with clustering and mobile sink technology. We first divide the whole sensor area into several sectors with the same size. In each cluster, a CH is selected according to the weight, which is calculated using the residual energy and the distance between the source node and the CH. Source nodes communicate with the CH using single or multi-hop communication in accordance with the optimal energy consumption. Additionally, CHs are connected by a reasonable chain, which is constructed by the greedy algorithm for intercluster communication. The CH that is closest to the sink is chosen as the leader to communicate with the sink. Then, the mobile sink moves along a predefined trajectory for data gathering. Extensive simulations have been performed to compare our presented schema with some similar work. Additionally, we further study the influence of some parameters on the network and enhance the performance of the network.

The rest of the paper is organized as follows. Section 2 discusses some classic routing protocols and some latest research achievement adopting mobile sink. Section 3 presents the system model, which contains the network and energy models. Section 4 describes our algorithm in detail. Large quantities of simulations are conducted for comparing our presented schema with some similar work and the results are discussed in Section 5. Section 6 analyzes the influence of different network parameters. Section 7 discusses some phenomena during the simulation and proposed some open research issues. Section 8 concludes this paper.

## 2. Related Work

During recent years, much attention has been paid to energy-efficient routing protocols. Energy-efficient cluster-based dynamic routing algorithm (ECDRA) [13] is a mobile sink-based routing schema. In ECDRA, the mobile sink is deployed at the outer side of the circular sensor field and rotates in a circular manner. The topology of the network changes dynamically according to the position of the mobile sink.

LEACH [15] is one of the most well-known and representative hierarchical routing protocols that was first proposed. In LEACH, all sensors are divided into two types, cluster heads (CHs) and ordinary nodes (ONs). An ON will deliver its monitored data to its corresponding CH, and the CH will fuse and forward the monitored data to the base station (BS). LEACH is much superior to traditional routing protocols in terms of extending the network lifetime. However, due to the random election of CHs, CHs are often unevenly distributed, and CHs communicate with the BS directly, causing large energy dissipation.

Power-efficient gathering in sensor information systems (PEGASIS) [16] is an enhanced version of LEACH, which is a chain-based hierarchical protocol. In PEGASIS, each node only needs to transmit the data package to its nearest neighbor, which is closer to the BS than the source node. CHs are connected into a chain by the greedy algorithm for intercluster communication. Then, each chain leader, which is closest to the BS, takes the responsibility to forward the data packages to the BS. The chain construction makes an economical use of energy by avoiding long-distance communication. Meanwhile, because of using multi-hop propagation, serious network delay could be caused. Therefore, this protocol is not suitable for delay-sensitive applications.

Hybrid, energy-efficient, distributed clustering approach (HEED) [17] is greatly improved alternative regarding aspects of CHs selection compared to LEACH. It uses the residual energy as a main parameter and nodes with higher energy have a bigger possibility to be the CHs. In this way, the time when the first node dies could be greatly postponed, and more uniform CHs could be generated. Intracluster communication cost is the secondary parameter to determine which cluster a node should take part in. Intracluster communication cost could be estimated using the average minimum reachability power (AMRP) as shown in Equation (1):(1)$$\mathrm{AMRP}=\frac{\sum_{i=1}^{M}MinPwr_{i}}{M}$$
where MinPwr denotes the minimal transmission power demanded by a node to communicate with its corresponding CH. Using AMRP further balances the energy consumption of the whole network. 

In the above routing protocols, the position of the sink is usually fixed, which results in the “hot spots” problem. With the popularity of the portable mobile devices, such as Radio Frequency Identification (RFID), smart phones, and Personal Digital Assistants (PDAs), the further application of the robots, Unmanned Ground Vehicles (UGVs), and Unmanned Aerial Vehicles (UAVs), routing protocols using mobile sinks have become a hot topic in recent several years. Below are some routing protocols using single mobile sink.

Zhao et al. [18] proposed an algorithm called Load Balanced Clustering and Dual Data Uploading (LBC-DDU). In LBC-DDU, the whole network is divided into three layers: sensor layer, cluster head layer, and SenCar layer. At the beginning of each round, the SenCar could calculate the optimized path in advance and then walk along the path to gather information utilizing single-hop transmissions. After visiting each selected point, the SenCar will return to the base station and make preparations for the next round. The SenCar is equipped with two antennas such that it can exchange information with two CH simultaneously utilizing the technology of Multi-User Multiple-Input and Multiple-Output (MU-MIMO), which makes contributions toward delay reduction and efficiency improvement.

Zhu et al. [19] proposed a tree-cluster-based data-gathering algorithm (TCBDGA). In TCBDGA, the weight of each node is calculated using several main factors including residual energy, the number of neighbors and the distance to the BS. Each node chooses its neighbor with maximum weight as its parent node. In this way, a tree-construction is set up and then each tree is decomposed into several sub-trees according to its depth and its traffic load. Simulation results show that it has better performance in terms of energy consumption. 

Xie and Pan [20] introduced the mobile sink in WSNs with obstacles, where a spanning graphs-based scheduling mechanism was proposed. On the basis of this scheduling mechanism, the authors propose a heuristic path planning algorithm to find a shortest path to avoid the obstacles. A mobile sink is set to walk along the path and collect data from CHs via direct transmission. The mobile sink returns to the origin after data collection. Simulation results demonstrate that their proposed algorithm can extend network lifetime effectively. The scheduling mechanism they proposed makes a good contribution to reduce the complexity traversing in WSNs with obstacles.

Velmani and Kaarthick [21] proposed a velocity energy-efficient and link-aware cluster-tree (VELCT) scheme. This scheme mainly contains two phases, set-up phase and steady-state phase. In the set-up phase, CHs are selected based on the threshold value and clusters are formed with better performance via intra-cluster communication. After that, a few nodes are selected to serve as data collection nodes (DCNs) to construct a data collection tree (DCT). In the steady-state phase, CHs collect monitored data from its members and then forward the aggregated data to DCN. Experimental results show that their proposed scheme has better performance in network throughput, energy consumption and network latency.

Selvakumar and Swamynathan [22] presented an efficient data aggregation algorithm called cluster-chain mobile agent routing (CCMAR). In CCMAR, the cluster head selection value (CHSV) is utilized to choose the optimal CHs. Cluster members form chains to transmit their data to the sink. The residual energy level, signal strength and path loss are considered to schedule an optimal routing path for the mobile agent (MA).

Furthermore, some literature tries to optimize the performance of the network in aspects of energy consumption and lifetime by combining the particle swarm optimization algorithm [23,24,25], immune algorithm [26], bio-inspired ant colony optimization algorithm [27], clustering algorithms, etc.

Some researchers introduce multiple mobile sinks in WSNs and further improve the performance of the network. Tashtarian et al. [28] presented an event-driven algorithm using multiple mobile sinks. In the algorithm, each node has two states: monitoring and transmitting states. When an event is captured, the related node turns its state from monitoring to transmitting and a group of active nodes are generated to send packages to the mobile sink. In large scale WSNs, due to the limit of the mobility of the mobile sink, the network is divided into several parts and each part introduces a mobile sink. Wang et al. [29] presented a distance-aware routing algorithm using multiple mobile sinks to decrease the energy consumption of the network. Wang et al. [30] introduced multiple mobile sinks to move along the predetermined paths to gather raw data.

With the aim of prolonging the lifetime of the whole network, Banerjee et al. [31] introduced multiple mobile CHs with rich energy. The mobile CHs work in a collaborative manner to collect information from different sections of the sensor field. The base station takes responsibility for communicating with the mobile CHs. Three different strategies were adopted to reduce multi-hop communication and enhance the lifetime of the network. 

## 3. System Model

### 3.1. Basic Assumptions

In this paper, we consider that the sensing field is a circular region. Each node owns a unique ID that differs from others’ nodes. We make some basic assumptions as follows:All sensors are randomly deployed by vehicles, such as a plane, and remain stationary after deployment.All sensors have the knowledge of the location of the other nodes via the information exchange.All sensors have the same initial energy and their batteries cannot be changed. Once a sensor exhausts its energy, it will be useless.The transmission power of sensors could be adjusted based on the communication distance.The moving trajectory of the mobile sink is well-scheduled and it owns unlimited energy and communication range.

### 3.2. Network Model

In this paper, we deploy *N* sensors in a circular field with radius *R*. The *N* sensors are denoted as: {*N1, N2, …, Nn*}. We arrange the mobile sink within the sensor field and it can move freely in the sensor field. The whole sensor area is evenly divided into several sectors and each sensor belongs to a sector based on its geographical position, as is shown in Figure 1.

### 3.3. Energy Model 

Here, we adopt the energy model that is applied in literature [32]. As is shown in Figure 2, energy consumption mainly contains two parts, transmission consumption and reception consumption. The transmission part contains the signal generation and its enhancement. Once the signal is created, it will be enhanced by the amplifier. The amplifier uses two different powers to strengthen the signal according to the transmission distance. Therefore, the energy model for transmission is also divided into a free space model, which is for short-haul communication, and multi-path fading model, which is for long distance communication.

Transmission consumption represents the energy the transmission circuit and the amplifier use, and the reception consumption represents the energy the reception circuit uses. Each sensor will consume $E_{Tx}$ energy to transmit an *l*-bit data package over the communication distance of *d*, as is shown in Equation (2):(2)$$E_{Tx}\left(l,d\right)=\{\begin{array}{c} l\cdot E_{elec}+l\cdot \varepsilon_{fs}\cdot d^{2}\;\;\;\;\;if\;\;d<d_{0}\;\\ l\cdot E_{elec}+l\cdot \varepsilon_{mp}\cdot d^{4}\;\;\;\;\;if\;\;d\geq d_{0} \end{array}$$
where $E_{elec}$ denotes the energy consumption to run the transmitter or receiver circuit. $\varepsilon_{fs}$ and $\varepsilon_{mp}$ denote the amplification coefficient for the free space model and the multi-path fading model, respectively. $d_{0}$ is a threshold value and it can be calculated using:(3)$$d_{0}=\sqrt{\frac{\varepsilon_{fs}}{\varepsilon_{mp}}}$$

Reception consumption represents the energy usage of the reception circuit. The energy consumption to receive an *l*-bit data package can be calculated using Equation (4):(4)$$E_{Rx}\left(l\right)=l\cdot E_{elec}$$

## 4. Our Proposed Routing Schema

### 4.1. Clustering Formation

We partition the whole sensor field into several sectors and the number of sectors is determined by the CHs. Each node joins one cluster according to its geographical information. The angle of each sector is shown as Equation (5):(5)$$A=\frac{2\pi }{N_{CHs}}$$
where $N_{CHs}$ denotes the number of CHs. Then, we calculate the cluster that each node should take part in using Equation (6):(6)$$CH_{id}=\{\begin{array}{ll} \frac{\arctan\left(\frac{x}{y}\right)}{A} & if\;\;x\geq 0\;and\;y\geq 0\\ \frac{\arctan\left(\frac{x}{y}\right)+\pi }{A} & if\;x<0\\ \frac{\arctan\left(\frac{x}{y}\right)+2\pi }{A} & if\;\;\;x\geq 0\;and\;y<0 \end{array}$$
where *x* denotes the position of the sensor in coordinate axis *x* and *y* denotes the position of the sensor in coordinate axis *y.* We set the center of the sensor field as the origin of coordinates and the network after clustering is shown as Figure 3. 

### 4.2. CH Selection

We expect to select the nodes with as much energy as possible. Therefore, the residual energy of nodes is a significant metric in CH selection. Another important metric for CH selection is the distance between the CH and the sink. CHs take the responsibility of transmitting their members’ data to the sink and a great deal of energy is consumed. Therefore, it is necessary to select the nodes with as close a distance to the sink as possible. We use three different methods to calculate the weight of sensors and the result is discussed in the simulation. We used Equations (7)–(9) to calculate the weight of sensors:(7)$$W_{i}=E_{residual\_i}$$
(8)$$W_{i}=\frac{E_{residual\_i}}{Dis_{i\_to\_sink}}$$
(9)$$W_{i}=\frac{E_{residual\_i}{}^{2}}{Dis_{i\_to\_sink}}$$

The node closest to the sink will first broadcast its weight to the other members in the cluster and claim itself to be a tentative CH. Only nodes with the higher weight will broadcast their own weight and become a new CH candidate. Finally, the node with the maximal weight will be elected as the final CH in each cluster. The network after CH selection is shown as Figure 4.

Many researches regarding LEACH, TTDD, and CCMAR conduct the selection of CHs during each round. However, frequent CH selection will greatly increase the control messages such that much energy is wasted. In our presented schema, the CHs selection only happens when any of the CHs do not satisfy the condition of maximal weight. During each round, each member node will upload a data package that contains its weight for selection to its corresponding CH. When CHs find that their member nodes have higher weight than themselves, a re-clustering message will be broadcast in those clusters by the CHs.

### 4.3. Intracluster Communication

Long distance communication usually consumes a great deal of energy and shortens the lifetime of the network. Therefore, it is necessary to avoid long-distance communication during the intracluster communication. In our presented schema, member nodes can calculate the cost of energy consumption of different routing paths to select an optimal relay node or transmit data directly. As Lambrou et al. [33] discusses, the energy consumption of transmission is about 1000 times that of calculation. Therefore, it is wise for members to choose a better routing path via calculation. The energy consumption in the routing path using direct communication can be calculated as:(10)$$E_{1}\left(S_{i},CH_{Si}\right)=\{\begin{array}{c} l\cdot E_{elec}+l\cdot \varepsilon_{fs}\cdot d{\left(S_{i},CH_{Si}\right)}^{2}\;\;\;if\;d\left(S_{i},CH_{Si}\right)<d_{0}\\ l\cdot E_{elec}+l\cdot \varepsilon_{mp}\cdot d{\left(S_{i},CH_{Si}\right)}^{4}\;\;if\;d\left(S_{i},CH_{Si}\right)\geq d_{0} \end{array}$$
where $d\left(S_{i},CH_{Si}\right)$ denotes the distance between node *i* and its corresponding CH. $E_{1}\left(S_{i},CH_{Si}\right)$ or $E_{1}\left(S_{i},S_{j}\right)$ represents the direct communication between the source and target nodes. If the source node is far away from its CH, a relay node *j* will be chosen to forward the data package. The total energy consumption for the whole routing can be calculated using:(11)$$\begin{array}{cc} E_{2}\left(S_{i},S_{j},CH_{Si}\right) & =E_{Tx}\left(l,d\left(S_{i},S_{j}\right)\right)+E_{Rx}\left(l\right)+R_{Tx}\left(l,d\left(S_{j},CH_{Si}\right)\right)\\ & =3l\cdot E_{elec}+\varepsilon \cdot d^{2}\left(S_{i},S_{j}\right)+\varepsilon \cdot d^{2}\left(S_{j},CH_{Si}\right) \end{array}$$

Nodes compare $E_{1}\left(S_{i},CH_{Si}\right)$ with $E_{2}\left(S_{i},S_{j},CH_{Si}\right)$ and choose the lower one as the routing path. The intracluster communication can be determined using:(12)$$E\left(S_{i}\right)=\min\left(E_{1}\left(S_{i},CH_{Si}\right),E_{2}\left(S_{i},S_{j},CH_{Si}\right)\right)$$

### 4.4. Intercluster Communication

In this section, we use a greedy algorithm to construct a chain for intercluster communication to avoid long-distance communication. The process of the formation of the chain can be divided into the following steps:

*Step1*: The sink broadcasts a Chain_Formation message in the whole network to require all the CHs to report their ID and position.

*Step2*: Once the sink receives the information regarding CHs, it will choose the closest CH as the leader. The leader takes the responsibility to transmit the data to the sink directly.

*Step3*: The greedy algorithm is executed in other CHs and each CH only transmits its data to a relay CH that is closer to the sink compared with itself.

The result of chain generation is shown in Figure 5.

### 4.5. Migration Strategy of the Mobile Sink

In our proposed schema, the moving trajectory of the mobile sink is a circle, as Figure 6 demonstrates. The mobile sink moves with a constant angular velocity and the current position of the mobile sink can be calculated using its initial position and moving time. Hence, the mobile sink only needs to announce its initial position and angular velocity such that the broadcast message of the network is greatly reduced. We suppose the initial position of the mobile sink is $P_{0}$, and after a Δt time interval, the mobile sink will have moved to a new position $P_{\Delta t}$, as is shown in Figure 6. The prediction of the mobile sink demands that the clock of sensors is synchronized and the position information will be calibrated every few rounds. When there is only one mobile sink in the network, the initial position of the mobile sink $P_{0}$ can be set at any position of the circle with radius *r*. Once there are multiple mobile sinks, their initial positions are evenly distributed at the edge of the circle with radius *r*.

## 5. Performance Evaluation 

### 5.1. Simulation Environment

In order to test our presented schema, we adopted the MATLAB with the Chinese version of R2016a simulation environment. Some relevant parameters are shown as Table 1.

### 5.2. Comparation of Different Algorithms

We compared our presented schema with CCMAR and ECDRA. In order to equally evaluate the performance of different algorithms, we adopted the same network model to execute the algorithms. As is commonly discussed in similar work, the network lifetime is commonly defined as the time when the first dead node appears in the network. We compared the network lifetime of different algorithms and the simulation result is shown as Figure 7. From Figure 7, we can clearly see that the schema we present had a better performance in the aspect of network lifetime compared to CCMAR and ECDRA. CCMAR adopts chain structure for intracluster transmission, which causes the heavy burden of forwarding. Additionally, intercluster communication in CCMAR adopts direct communication between CHs and the mobile sink, which greatly increases the energy consumption of CHs. ECDRA also adopts a mobile sink to rotate in the sensor field; however, it only considered the mobile sink moving in the outer side of the sensor field. Therefore, sensors in the boundary dissipate more energy and that hastens their demise. Our proposed schema performed best because it calculated the energy for different routing paths and chooses the optimal one.

Then, we compared the energy consumption between different algorithms and the result is illustrated as Figure 8. We can clearly see from Figure 8 that our presented schema still had the best performance. Too much data forwarding resulted in the unnecessary energy dissipation in CCMAR and that led to a higher energy consumption compared to the other two algorithms. ECDRA consumed more energy than our presented algorithm because the average communication distance in ECDRA was much longer than that in our presented schema.

## 6. Network Parameters Adjustment and Performance Enhancement

In this section, we discuss the influence of different parameters on the performance of the network. The radius of the moving trajectory of the mobile sink has a significant influence on the topology construction. Then, the method for weight calculation determines the usage ratio of the energy. The number of the clusters and the speed of the sink may also affect the performance. Additionally, multiple mobile sinks are also discussed in this section.

### 6.1. Study of the Radius of the Moving Trajectory

In our proposed schema, the mobile sink rotates in a circle and the radius of the circle needs to be optimized. We first changed the radius of the moving trajectory and kept other parameters unchanged. We set the radius to 0, 0.25R, 0.5R, 0.75R, and R, and the lifetime of the network is shown as Figure 9. From Figure 9, we found that when the mobile sink moved along the radius of 0.25R, the network achieved the longest lifetime. However, when the mobile sink ran at the outside of the circle, the network had the poorest performance. 

We compared the energy consumption of different radii for single mobile sink. Simulation results are shown in Figure 10. We can obviously see that the energy consumption rose much slower when the radius of the mobile sink was about 0.25R. Furthermore, when the radius of the mobile sink was about R, the energy consumption of the network rose rapidly. The reason for the change could be that we took residual energy of each node into consideration such that the energy consumption of the whole network was not optimal.

Additionally, we studied the optimal radius of the moving trajectory under the different network sizes and we set the network size as 100, 200, 300, and 400 meters. With the network size R increasing, the whole energy consumption increased and the lifetime of the network decreased. As is shown in Figure 11, 0.25R was still the optimal radius for the mobile sink under the different network scales.

### 6.2. Study of the Methods of Weight Calculation of CHs

It is a good choice to select sensors with high residual energy as CHs. However, those nodes with high residual energy tend to be very far away from the mobile sink. Consequently, if we only considered the residual energy of nodes, long distance communication will occur between CHs and the sink. Thus, it was necessary to achieve a balance between the residual energy and distance from CHs to mobile sink to prolong the lifetime of the system. We first set the radius of the mobile sink to be 0.25R, and then we calculated the weight of each node by using Equations (7)–(9). Simulation results are provided in Figure 12, and it illustrates that it is more reasonable for us to use Equation (9) to calculate the weight of nodes. 

A great deal of research defines the network lifetime as the time when the first node dies. However, using Equation (8) may make the node’s death premature because the distance accounts for a large proportion in the weight calculation. In order to highlight the importance of the residual energy, we adopted Equation (9) as the method of weight calculation under the circumstances that the initial energy of each node was less than 1 J. Simulation results demonstrate that it performed better in terms of the time when the first node dies.

### 6.3. Study of the Number of Clusters

The number of clusters was another important parameter of the network. Too few CHs may cause long-distance communication between cluster members and CHs. However, excessive CHs caused oversized chains to be generated during intracluster communication. Plenty of energy will be wasted to forward the data packages and the network latency will greatly increase. We set different cluster numbers in our schema when the radius of the mobile sink was 0.25R, and the simulation results are shown in Figure 13. When the cluster number was 5, the network had a better performance, especially with regard to the time when the first node died.

### 6.4. Study of the Speed of the Mobile Sink 

We also studied the speed of the mobile sink. As the sink is carried by vehicles, such as the intelligent car, its speed is controllable. Therefore, we set the angular velocity of the mobile sink as π/20, π/10, and π/5, and the radius of the mobile sink was 0.25R. The simulation results are shown in Figure 14. It can be clearly seen that the lifetime of the network was almost unchanged with the increasing speed of the mobile sink. Therefore, we draw a conclusion that the speed of the mobile sink has little influence on the lifetime of the network, and it should not be too fast or too slow.

### 6.5. Study of Multiple Mobile Sinks 

We also studied the influence of multiple mobile sinks on the lifetime of the network. The moving radius of each sink was set to 0.25R. Simulation results illustrate that with the increasing numbers of mobile sinks, the lifetime of the network improved, as is shown in Figure 15. We also noticed a trend that with the number of mobile sinks increasing, the improvement of the lifetime becomes fewer and fewer. Since the mobile sinks are more expensive than ordinal sensors, three mobile sinks are recommended to be the most cost-effective.

We compared the energy consumption of multiple mobile sinks when the radius of the mobile sink was 0.25R. It can clearly be seen from Figure 16 that when the number of the mobile sinks increased, the upward trend of the energy consumption of the whole network decreased. When the number of the mobile sinks exceeded three, the performance of the whole network does not change much.

## 7. Discussions and Open Research Issues

### 7.1. Uneven Energy Distribution Between Clusters 

The results show that our proposed algorithm had good performance in terms of its lifetime; however, we found a common phenomenon that the energy distribution between clusters was uneven. As is shown in the Figure 17, some clusters had no node alive, whereas the nodes in some clusters were all alive. The following reasons may cause the problem:Sensors were deployed unevenly and each cluster could have different number of nodes.The average distance between nodes and the regional center in one cluster could be longer than that in other clusters. 

Node deployment had a great influence on the performance of the network. The study of node deployment strategies has been a hot research issue in recent years since it can improve the performance of WSNs in the aspects of energy consumption, network lifetime, and coverage rate. Homogeneous deployment of sensor nodes, if possible, can eliminate the phenomenon mentioned above.

### 7.2. Open Research Issues

It has been proven that the optimal sink moving trajectory problem is an NP-hard issue [34,35] that is, by definition, hard to solve. More importantly, the sink mobility issue is tightly related with routing algorithms used therein. Thus, jointly optimizing mobile sink strategies and routing algorithms is one of the primary research challenges. Not only does a proper moving trajectory, cluster head selection, sink number, speed, moving pattern, and sojourn positions need to be carefully designed, but also the corresponding routing algorithms need to be selected for different applications.

Security is always an important and challenging research issue for WSNs. Restricted to the limitation of resources and deployment environment, it is difficult to implement strong security algorithms [36]. Besides, due to the potential threatens like data modification and malicious attacks, communications between source and destination nodes may have serious influence on the network security and increase the energy consumption during data transmission. Using sink mobility technology in WSNs, various types of data reports may get lost or modified during data transmission due to the change of the location of the sink and selective forwarding. Therefore, security as an essential component for data transmission in dynamic sensor networks environment needs further research attention. It is likely that the security for store and forward [37] will help in this regard.

In many location-based applications, data obfuscation technologies can be used to protect location privacy to avoid the damage caused by having the position leaked. By reducing location accuracy, data obfuscation technology can achieve the fuzzy effect of the locations of multiple data information. However, the safety of this technology remains to be further researched. For example, a general three-tier security framework has been proposed [38] where two separate key pools are respectively designed for a mobile sink node and pair-wise key establishment between sensor nodes. This scheme can effectively protect sensor networks from stationary access node replication attacks.

## 8. Conclusions

Clustering and sink mobility technology as effective methods to improve the performance of WSNs have drawn much research attention recently. In this paper, we first divided the whole sensor field into several sectors with the same size, and each sector selected a CH according to its members’ weight. Cluster members calculated the routing path with the optimal energy consumption to transmit data to their corresponding CHs via single or multi hops communication. Then, the CHs adopted a greedy algorithm to form a chain for intercluster communication. The latest routing schemas, such as CCMAR and ECDRA, were compared with our presented algorithm regarding aspects of energy consumption and network lifetime by simulation. The simulation results proved our presented algorithm outperformed the other two algorithms. Additionally, we explored the influence of different parameters on the performance of the network and further improved its performance. 

## Figures and Tables

**Figure 1 sensors-19-01494-f001:**
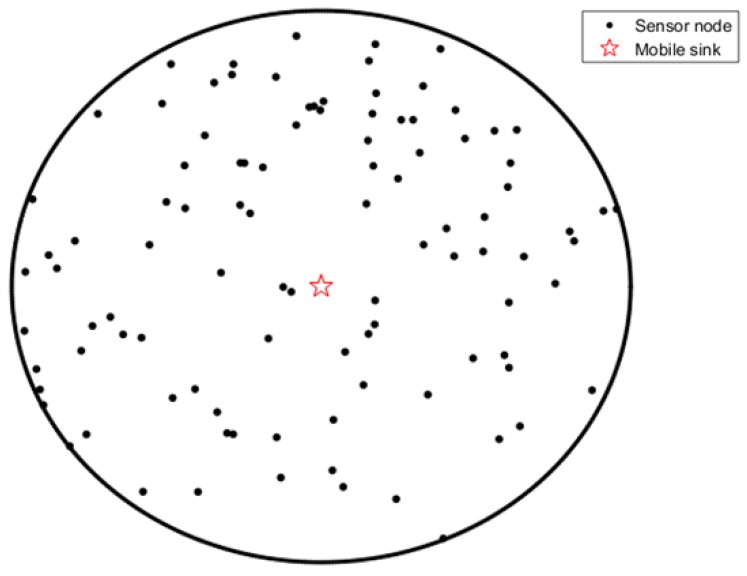
Network model.

**Figure 2 sensors-19-01494-f002:**
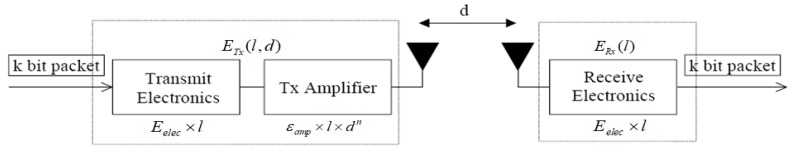
Energy model.

**Figure 3 sensors-19-01494-f003:**
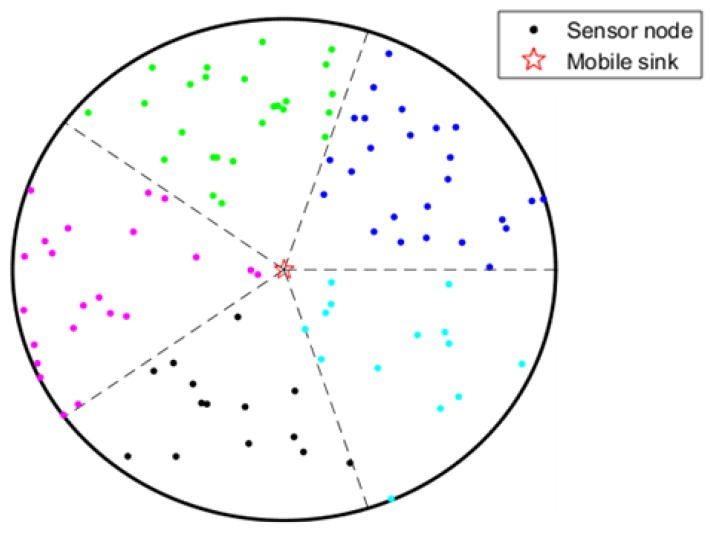
Clustering formation.

**Figure 4 sensors-19-01494-f004:**
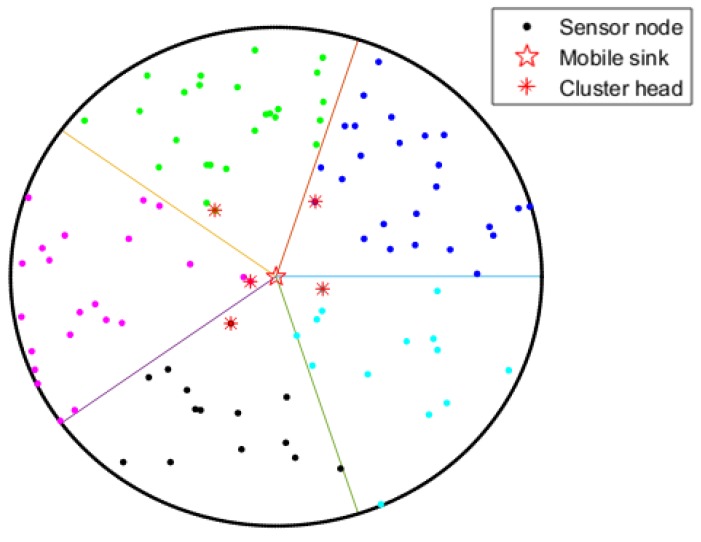
CHs selection.

**Figure 5 sensors-19-01494-f005:**
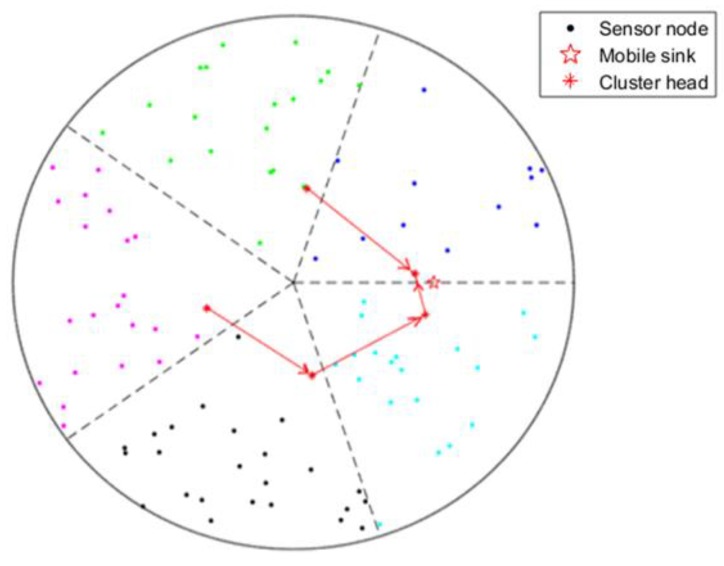
Chain of CHs generation.

**Figure 6 sensors-19-01494-f006:**
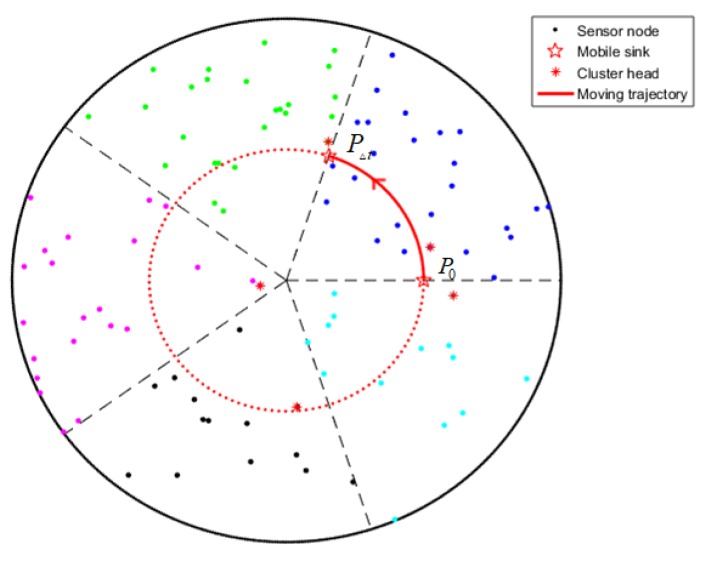
Trajectory of mobile sink.

**Figure 7 sensors-19-01494-f007:**
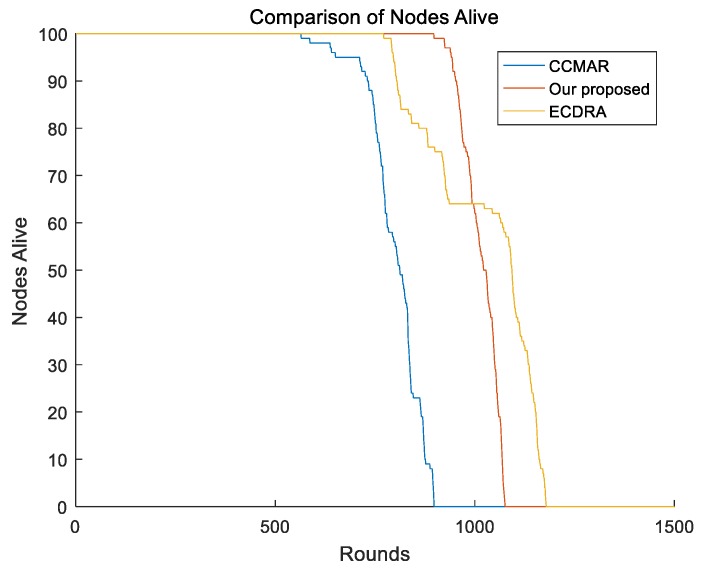
Comparison of the network lifetime between different algorithms.

**Figure 8 sensors-19-01494-f008:**
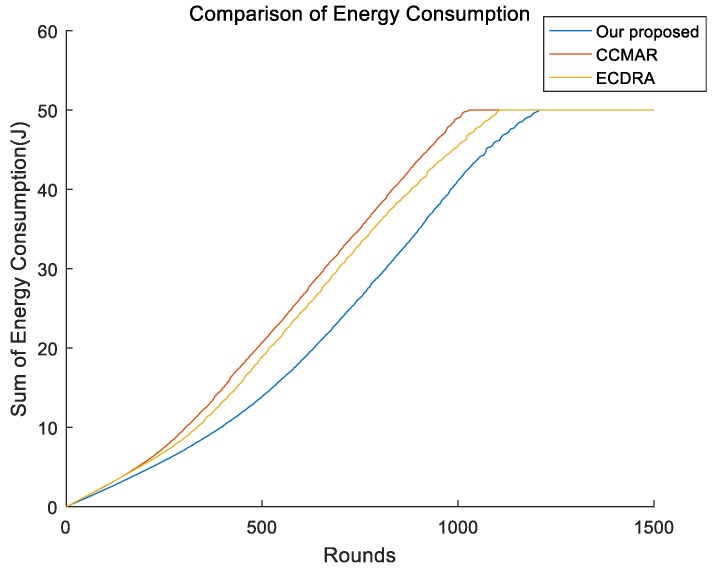
Comparison of total energy consumption between different algorithms.

**Figure 9 sensors-19-01494-f009:**
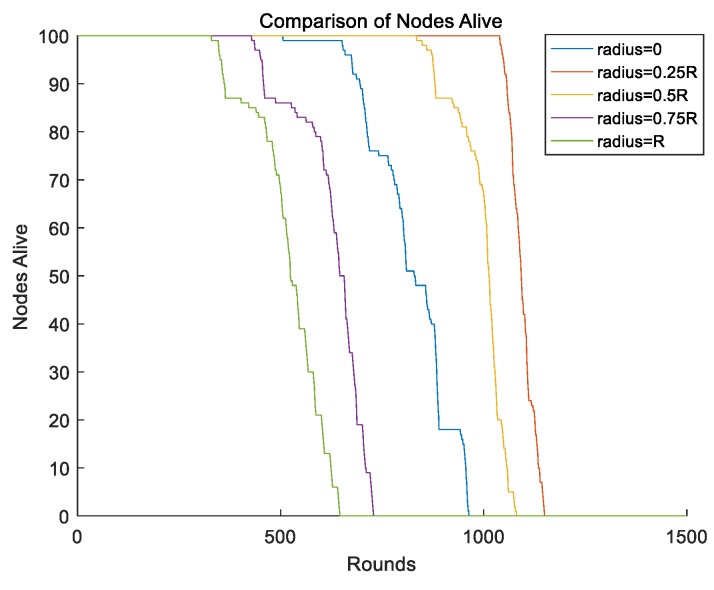
Different radius of mobile sink.

**Figure 10 sensors-19-01494-f010:**
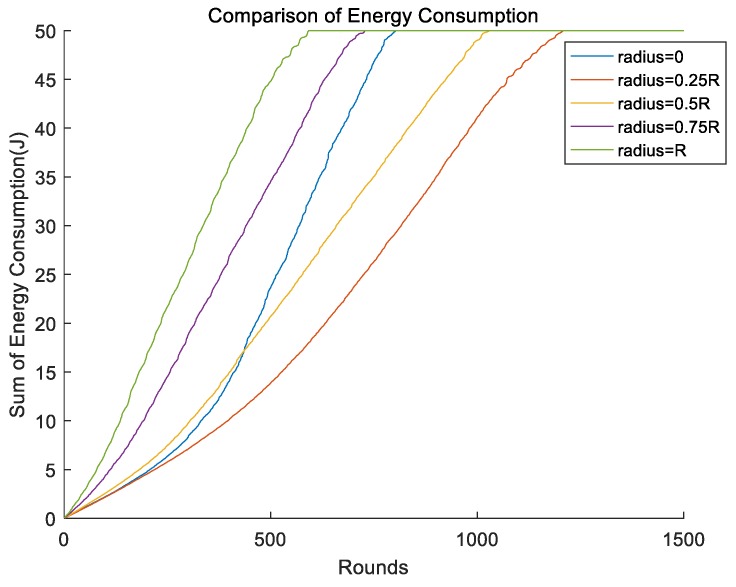
Energy consumption of different radius of single mobile sink.

**Figure 11 sensors-19-01494-f011:**
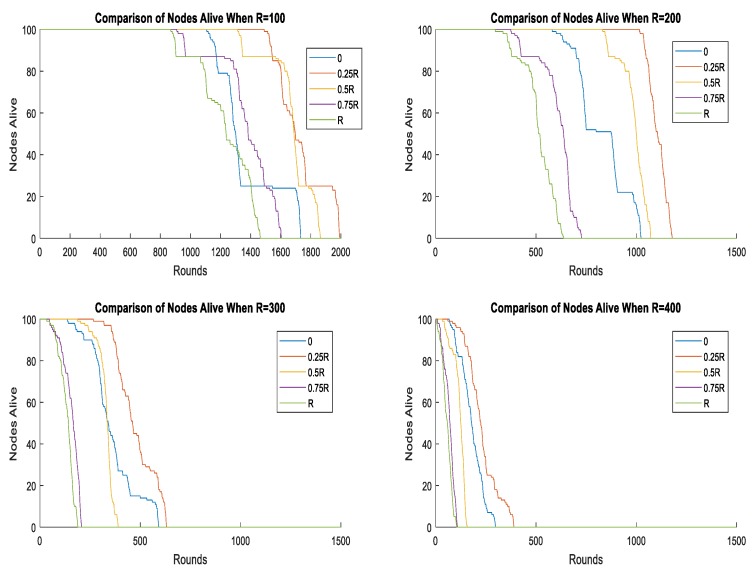
Different radius of mobile sink under different network size.

**Figure 12 sensors-19-01494-f012:**
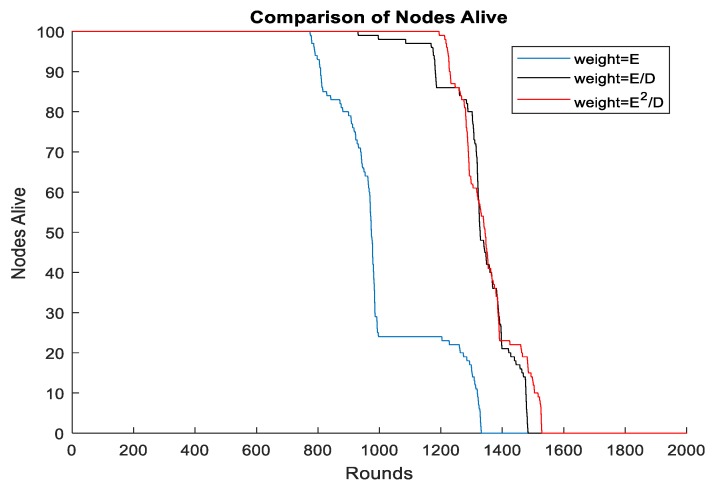
Different methods of weight calculation.

**Figure 13 sensors-19-01494-f013:**
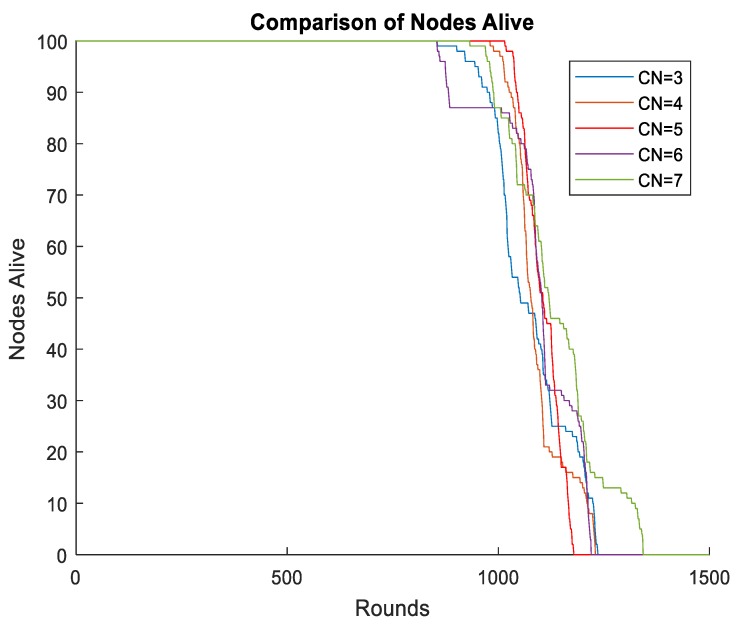
Different number of clusters.

**Figure 14 sensors-19-01494-f014:**
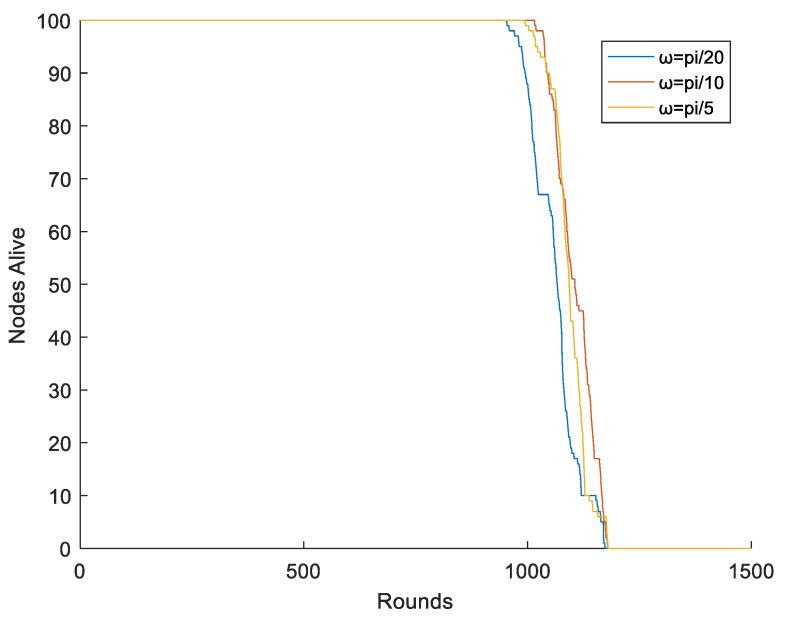
Different speed of the mobile sink.

**Figure 15 sensors-19-01494-f015:**
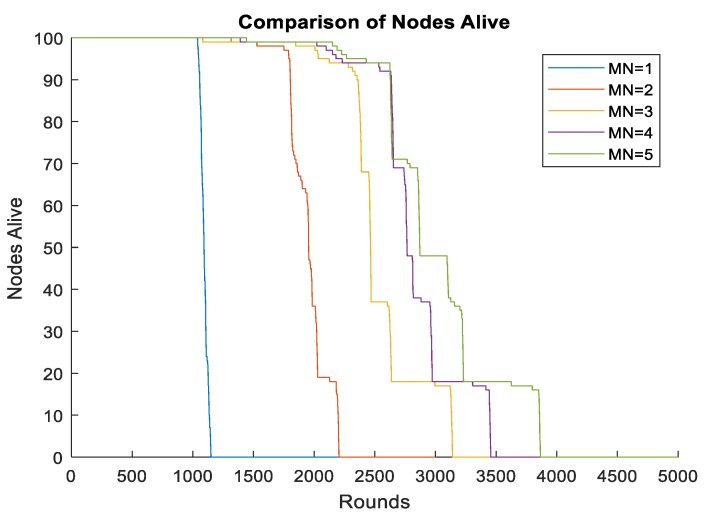
Different mobile sink number.

**Figure 16 sensors-19-01494-f016:**
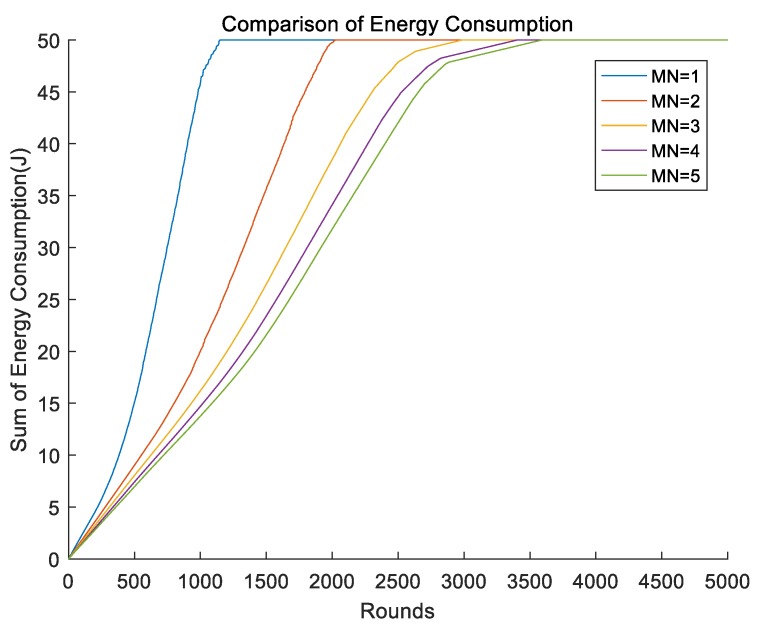
Energy consumption of different numbers of mobile sinks.

**Figure 17 sensors-19-01494-f017:**
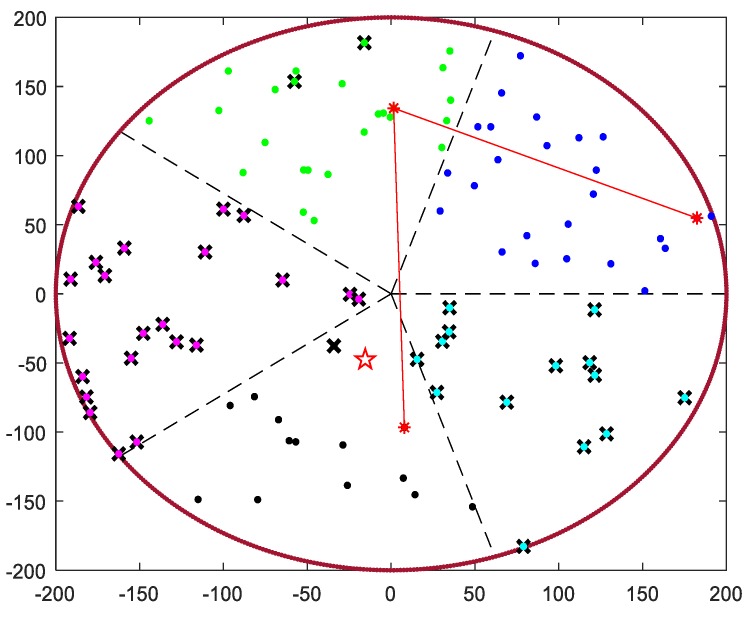
Uneven energy distribution between clusters.

**Table 1 sensors-19-01494-t001:** Simulation parameters.

Parameter Name	Value
Network radius (R)	[100, 200, 300, 400] m
Mobile sink radius (r)	[0, 0.25R, 0.5R, 0.75R, R]
Mobile sink number (MN)	[1, 2, 3, 4, 5]
Mobile sink speed (w)	[π/20, π/10, π/5]
Cluster number (CN)	[3, 4, 5, 6, 7]
Number of nodes (N)	100
Packet length (l)	500 bits
Initial energy ($E_{0}$)	0.5 J
Energy consumption on circuit ($E_{elec}$)	50 nJ/bit
Free-space model parameter ($\varepsilon_{fs}$)	10 pJ/bit/m^2^
Multi-path model parameter ($\varepsilon_{mp}$)	0.0013 pJ/bit/m^4^
Distance threshold ($d_{o}$)	$\sqrt{\frac{\varepsilon_{fs}}{\varepsilon_{mp}}}$m

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
