# Peer review of "Energy Efficient Routing Algorithm with Mobile Sink Support for Wireless Sensor Networks"

_sensors, 2019, doi:10.3390/s19071494_

Round 1
Reviewer 1 Report
Spelling, Grammar and Style
Please check each sentence for typographical error and grammar, especially for the following line numbers:
48 – it’s and its
61 - rephrase
245 – select instead of selected
249 – 250. Three methods, instead of two, use instead of using
310 – Comparison , remove period on the subheading.
313 - rephrase
435 - rephrase
276 - E1(Si, Sj) or E1 (Si, CH) referring to the direct communication energy consumption cost.
Technical, Originality and Contribution
Is there a basis to use the mobile sink to rotate in a circular manner? Is this kind of assumption the most efficient mobile sink movement or just an assumption for simplicity of simulation? The paper Title indicates of Mobile Sink Support but this does not reflect on the content as the movement is set to a circular manner, scheduled and predicted. There is no clear addition, contribution or novelty in the way the mobile sink has been implemented, rather most of it is assumptions.
There is no definite mention of the initial position of (P0) the mobile sink in the simulation, it is assumed to be placed based on the Mobile Sink Radius Parameter which has 0 (center) and successively going towards the outside of the network radius (0.25 – R). With these, it can be assumed that with using multiple mobile sink, there is a possibility that 2 or more sink is placed in one sector?
Since the mobile sink moves in a predefined way and sensor nodes are static, there would be multiple rounds that a CH is not relieved in the network. There should be mention if the CH Selection is done per round in the simulation as it can be assumed that CH selection only happens once, only at the deployment stage? Or it could be better to indicate the analysis when does the change in CH happens in using the proposed approach compare with the other schemes.
The paper has taken well into consideration different parameters for simulation that might affect the performance of the proposed scheme such as the radius of the trajectory of the mobile sink, network area size, speed, number of clusters, etc. However, comparison with LEACH which mostly has not been used for mobile sink might pose bias results. Comparison with several routing schemes that have considered mobile sink scenario might be needed.
Author Response
Response to comment1:
1. Spelling, Grammar and Style
Please check each sentence for typographical error and grammar, especially for the following line numbers:
48 – it’s and its
61 - rephrase
245 – select instead of selected
249 – 250. Three methods, instead of two, use instead of using
310 – Comparison , remove period on the subheading.
313 - rephrase
435 - rephrase
276 - E1(Si, Sj) or E1 (Si, CH) referring to the direct communication energy consumption cost.
Thanks for your detailed recommendations and we have carefully proofread each sentence especially you proposed to make sure that there are no text errors.
Technical, Originality and Contribution
2. Is there a basis to use the mobile sink to rotate in a circular manner? Is this kind of assumption the most efficient mobile sink movement or just an assumption for simplicity of simulation? The paper Title indicates of Mobile Sink Support but this does not reflect on the content as the movement is set to a circular manner, scheduled and predicted. There is no clear addition, contribution or novelty in the way the mobile sink has been implemented, rather most of it is assumptions.
Thanks, the controllable mobility schema can be divided into two categories, unrestricted schema and geographic restricted model. Many papers adopt rectangular sensor fields and use mobile sinks to gather information along a predefined regular path. However, the sensor field in some applications such as the buses running at the fixed path may be a rounded field, and it is reasonable to assume that the mobile sink rotates in a circular manner. In this manuscript, we mainly scheduled the moving route of the mobile sink by adjusting it’s moving radius as well as optimize the cluster number, the selection method of CHs and some other important factors. Regular moving schema may not be the most efficient approach of the mobile sink movement, however, it is still meaningful in some specific applications.
3. There is no definite mention of the initial position of (P0) the mobile sink in the simulation, it is assumed to be placed based on the Mobile Sink Radius Parameter which has 0 (center) and successively going towards the outside of the network radius (0.25 – R). With these, it can be assumed that with using multiple mobile sink, there is a possibility that 2 or more sink is placed in one sector?
Thanks for pointing out this. In our presented schema, the mobile sink owns a constant moving radius. If there is only one mobile sink, the initial position can be set at any point of the edge of the circle with radius r because it can traverse the whole edge of the circle after a few rounds at any initial point. Once there are multiple mobile sinks, the initial points of different mobile sinks will be evenly distributed at the edge of the circle with radius r. if the mobile sinks are much enough or the angle of the sector is big enough, it may occur that 2 or more mobile sinks are placed in one sector. However, we do not need so many mobile sinks and the angle of the sector is commonly not big. Therefore, it is impossible for two or more mobile sinks to be placed at one sector. We have also added the explanation in section 4.5.
4. Since the mobile sink moves in a predefined way and sensor nodes are static, there would be multiple rounds that a CH is not relieved in the network. There should be mention if the CH Selection is done per round in the simulation as it can be assumed that CH selection only happens once, only at the deployment stage? Or it could be better to indicate the analysis when does the change in CH happens in using the proposed approach compare with the other schemes.
Thanks for your valuable suggestion. Frequent CHs selection will greatly increase the control messages so that much energy is wasted. In our presented schema, the CHs selection only happens when any of the CHs doesn’t satisfy the condition of maximal weight. During each round, each member node will upload a data package to its corresponding CH and it contains its weight for selection. When CHs finds that their member nodes have higher weight than themselves, re-clustering message will be broadcast in those clusters by the CHs. We have also added explanation in section 4.2 and give some discussion between different schemas.
5. The paper has taken well into consideration different parameters for simulation that might affect the performance of the proposed scheme such as the radius of the trajectory of the mobile sink, network area size, speed, number of clusters, etc. However, comparison with LEACH which mostly has not been used for mobile sink might pose bias results. Comparison with several routing schemes that have considered mobile sink scenario might be needed.
Thanks for your valuable comment. The result of the comparison between LEACH and our presented schema is obvious because sink mobility will greatly increase the performance of the network. In order to have a more equal comparison, we compare our presented schema with CCMAR (2017) and ECDRA (2017) which adopt mobile sinks. We extend our simulation and have a detailed discussion in section 5.2.
Reviewer 2 Report
The paper is well written and very clear, but it contains some spelling mistakes and typos that I will detail at the end of this report. They should be addressed before publication.
Regarding the content, the objectives of the paper are clear and connected with previous research in a very concise way, which I appreciate. Interesting evaluation results are presented and compared with existing state-of-the-art methods.
I would make only one recommendation. The movement of the mobile sink is not clearly described until section 6.1, where it says that "the mobile sink rotates in a circle and the radius of the circle..." Before that, it is only said that the mobile sink moves in a regular way and that the position could be derived from the initial position and the speed, which was puzzling for me (how can one know the position without knowing the direction?). So, please indicate much earlier (probably in the Introduction, but also in 4.5, for sure) that the movement of the sink is circular.
One more thing related to the circular movement. In the study of the speed in subsection 6.4: what is the Radius used for this calculation? The translation from angular speed to linear speed is only possible if the radius is know. Are the values given for a fixed radius 0.25R because of the study reported in Figure 11? Also, why did the authors decide to choose based on angular speed when the limiting factor is the linear speed of the vehicles? I suggest to transform the speed into linear units (either m/s or kmh) as this is the usual metrics for vehicle speed.
TYPOS:
Replace it's with "it is"; it appears in a few sentences, including the abstract.
In page 3 there is an equation defining AMRP; note that the first member of the equation is formatted as four different letters A M R P, please fix it.
The term literature is plural as it denotes a set of papers. Although the plural "literatures" can be used in certain contexts, it is not correctly used in this paper. Please replace any instance of "literatures" with the singular:
In the literatures -> in the literature
Author Response
Response to comment1:
1. I would make only one recommendation. The movement of the mobile sink is not clearly described until section 6.1, where it says that "the mobile sink rotates in a circle and the radius of the circle..." Before that, it is only said that the mobile sink moves in a regular way and that the position could be derived from the initial position and the speed, which was puzzling for me (how can one know the position without knowing the direction?). So, please indicate much earlier (probably in the Introduction, but also in 4.5, for sure) that the movement of the sink is circular.
Thanks for pointing this out. The speed of mobile sink denotes the angular velocity. The moving trajectory of the mobile sink is a scheduled circle and the angular velocity is constant. Therefore, we can infer the current position of the mobile sink according to its initial position and moving time. We have added a detailed description on the movement of the mobile sink in section 4.5 and repaint Figure 6 for a better illustration.
2. One more thing related to the circular movement. In the study of the speed in subsection 6.4: what is the Radius used for this calculation? The translation from angular speed to linear speed is only possible if the radius is know. Are the values given for a fixed radius 0.25R because of the study reported in Figure 11? Also, why did the authors decide to choose based on angular speed when the limiting factor is the linear speed of the vehicles? I suggest to transform the speed into linear units (either m/s or kmh) as this is the usual metrics for vehicle speed.
Thanks for your suggestion. The Radius used for section 6.2, 6.3, 6.4 and 6.5 is 0.25R and we have added the explanations in those sections respectively. We use angular velocity just because it is convenient for simulation. Angular velocity is more useful when calculating the current position of the mobile sink.
TYPOS:
3. Replace it's with "it is"; it appears in a few sentences, including the abstract.
Thanks, we have revised it.
4. In page 3 there is an equation defining AMRP; note that the first member of the equation is formatted as four different letters A M R P, please fix it.
Thanks, we have adjusted the size of the equation to fix the problem.
5. The term literature is plural as it denotes a set of papers. Although the plural "literatures" can be used in certain contexts, it is not correctly used in this paper. Please replace any instance of "literatures" with the singular:
In the literatures -> in the literature
Thanks, we have replaced all the “literatures” with “literature”.
Round 2
Reviewer 1 Report
All the inquiries and comments with the previous review were clarified and added.
Minor checking of the content and structure should be done.
The compared ECDRA scheme was not mentioned in the conclusion.
Author Response
Response to comment 1:
1. Minor checking of the content and structure should be done.
Yes, we have rechecked the whole paper carefully for text and structure errors.
2. The compared ECDRA scheme was not mentioned in the conclusion.
Many thanks, we have revised the relevant sentence by adding comparison with ECDRA scheme in the conclusion.